# A microRNA Profile Regulates Inflammation-Related Signaling Pathways in Young Women with Locally Advanced Cervical Cancer

**DOI:** 10.3390/cells13110896

**Published:** 2024-05-23

**Authors:** Oliver Millan-Catalan, Eloy Andrés Pérez-Yépez, Antonio Daniel Martínez-Gutiérrez, Miguel Rodríguez-Morales, Eduardo López-Urrutia, Jaime Coronel-Martínez, David Cantú de León, Nadia Jacobo-Herrera, Oscar Peralta-Zaragoza, César López-Camarillo, Mauricio Rodríguez-Dorantes, Carlos Pérez-Plasencia

**Affiliations:** 1Laboratorio de Genómica, Instituto Nacional de Cancerología, Tlalpan, Mexico City 14080, Mexico; oliver.millan.sg@gmail.com (O.M.-C.); eperezy2306@gmail.com (E.A.P.-Y.); maga94@comunidad.unam.mx (A.D.M.-G.); 2Posgrado en Ciencias Biológicas, Unidad de Posgrados, Universidad Nacional Autónoma de México (UNAM), Ciudad Universitaria, Coyoacán, Mexico City 04510, Mexico; 3Laboratorio de Genómica, FES-Iztacala, Universidad Nacional Autónoma de México (UNAM), Iztacala, Tlalnepantla 54090, Mexico; leugimroselarom@gmail.com (M.R.-M.); e_urrutia@unam.mx (E.L.-U.); 4Unidad de Investigaciones Biomédicas en Cáncer, Instituto Nacional de Cancerología, Tlalpan, Mexico City 14080, Mexico; quiechc8@hotmail.com (J.C.-M.); dfcantu@gmail.com (D.C.d.L.); 5Unidad de Bioquímica, Instituto Nacional de Ciencias Médicas y Nutrición Salvador Zubirán, Tlalpan, Mexico City 14080, Mexico; 6Dirección de Infecciones Crónicas y Cáncer, Centro de Investigación Sobre Enfermedades Infecciosas, Instituto Nacional de Salud Pública, Cuernavaca, Morelos 62100, Mexico; operalta@insp.mx; 7Posgrado en Ciencias Genómicas, Universidad Autónoma de la Ciudad de México, Mexico City 03100, Mexico; cesar.lopez@uacm.edu.mx; 8Laboratorio de Oncogenómica, Instituto Nacional de Medicina Genómica, Tlalpan, Mexico City 14610, Mexico; mrodriguez@inmegen.gob.mx

**Keywords:** miRNAs, locally advanced cervical cancer, JAK/STAT signaling pathway, inflammation, young female cancer patients

## Abstract

Cervical cancer (CC) remains among the most frequent cancers worldwide despite advances in screening and the development of vaccines against human papillomavirus (HPV), involved in virtually all cases of CC. In mid-income countries, a substantial proportion of the cases are diagnosed in advanced stages, and around 40% of them are diagnosed in women under 49 years, just below the global median age. This suggests that members of this age group share common risk factors, such as chronic inflammation. In this work, we studied samples from 46 patients below 45 years old, searching for a miRNA profile regulating cancer pathways. We found 615 differentially expressed miRNAs between tumor samples and healthy tissues. Through bioinformatic analysis, we found that several of them targeted elements of the JAK/STAT pathway and other inflammation-related pathways. We validated the interactions of miR-30a and miR-34c with JAK1 and STAT3, respectively, through dual-luciferase and expression assays in cervical carcinoma-derived cell lines. Finally, through knockdown experiments, we observed that these miRNAs decreased viability and promoted proliferation in HeLa cells. This work contributes to understanding the mechanisms through which HPV regulates inflammation, in addition to its canonical oncogenic function, and brings attention to the JAK/STAT signaling pathway as a possible diagnostic marker for CC patients younger than 45 years. To our knowledge to date, there has been no previous description of a panel of miRNAs or even ncRNAs in young women with locally advanced cervical cancer.

## 1. Introduction

A total of 600,000 new cases of cervical cancer (CC) and 340,000 deaths caused by this disease are reported annually worldwide, which positions it as one of the most prevalent cancers among women [1]. In middle-income countries like Mexico, over 60% of patients are diagnosed in advanced stages, which remains a major public health problem [2,3]. The median age for diagnosis of CC patients is 50 years [4]; however, 42% are diagnosed at an earlier age [1]. In Mexico, 43% of CC patients are diagnosed between 18 and 49 years, mostly in advanced stages [2,5], and women with CC younger than 40 years represent 20% [6]. This suggests that women between 40 and 50 years old represent 23% of the population affected by this type of neoplasm. Therefore, we considered for this study patients younger than 45 years, who represent a unique group characterized by poor screening assessment and a wide range of clinical outcomes [6,7,8].

Human papillomavirus (HPV) infection, considered a principal risk factor for CC, leads to slow and complex cellular alterations in the cervix, driven by the HPV genomic integration and overexpression of the viral oncoproteins E6 and E7, which induces the repression of p53 and pRB tumor suppressor genes, respectively [9]. This molecular mechanism prompts cell cycle hyperactivation, DNA repair deregulation, and apoptosis evasion [10]. However, it has been demonstrated that persistent HPV infection is necessary but not sufficient for the development of CC [11,12]. Additional risk factors for the development of the disease include early onset of sexual activity, multiparity, bacterial vaginosis, sexually transmitted infections [13], and chronic inflammation [14,15].

The role of inflammation, especially chronic inflammation, in cancer development is receiving progressively more attention. The constant presence of molecular mediators of inflammation, such as interleukins and pro-inflammatory cytokines, in the tumor microenvironment is now widely accepted [14]. CC development is supported by inflammation-related signaling pathways such as MAPK, NF-kβ, COX-PG, and JAK/STAT [10,16]. The activation of the JAK/STAT pathway is particularly important as it is crucial for the regulation of cellular processes such as proliferation, survival, cell differentiation, and inflammation [17]; several of its components, such as cytokines, STAM1, JAK1/2, and STATs, participate in CC development and progression [18].

Inflammation in cervical cancer is further complicated by the presence of HPV, which stimulates pro-inflammatory pathways on its own [10] through mechanisms including regulating microRNAs (miRNAs)—non-coding, 20-nucleotide-long RNAs that negatively regulate mRNAs [19]. Various miRNAs, such as miR-199a-5p [20] or miR-9 [21], have been demonstrated to regulate the JAK/STAT pathway. However, the specific miRNAs regulating inflammation in CC patients remain to be determined.

In this work, we performed genomic profiling comparing miRNA expression levels in healthy tissues and cancer samples in a Mexican cohort of patients younger than 45 years with LACC. We identified 615 differentially expressed miRNAs in tumoral tissues; those expressions with higher fold changes and significant p-values were validated in an independent cohort of patients. As found through pathway enrichment analysis, the most represented targets were JAK/STAT signaling pathway elements. Then, we tested the direct regulation of JAK1 and STAT3 exerted by miR-30a and miR-34c, respectively, and validated its effect on the proliferation rates of the cervical cancer cell lines. The evidence presented in this work contributes to understanding the molecular mechanism that drives the progression of cervical cancer through miRNA regulation of the JAK/STAT pathway components.

## 2. Materials and Methods

### 2.1. Patient Cohort

A total of 46 women diagnosed with locally advanced cervical cancer (LACC) between 28 and 45 years old (considered as young women with cervical cancer) were included in this study. As a healthy reference, ten cervical tissue samples were obtained through hysterectomy of uterine myomatosis patients. All patients were treated in Instituto Nacional de Cancerologia (INCAN), Mexico; the patients included in this study accepted and signed an informed consent form according to protocol approved by the ethics and research committee of the institute (approval numbers 015/012/IBI CEI/961/15).

Inclusion criteria for patients were: (1) women with a confirmed pathologic diagnosis of CC in a locally advanced stage (IB2 up to IIIB) according to FIGO; (2) less than 45 years old; (3) biopsies with 80% of tumors cells confirmed by pathology report; (4) high-quality DNA and RNA; (5) no presence of comorbidities; (6) without previous oncological treatment; and (7) able to receive standard or conventional therapy based on concurrent CRT. Inclusion criteria for healthy tissues were: (1) donors under 45 years old, (2) no previous cervical surgery, (3) no HPV infection, (4) no hormonal treatment, and (5) at least three previous negative Pap smears.

### 2.2. RNA Purification

The RNA extraction was performed using two different techniques. First, for the microarray analysis, total RNA was extracted from fresh tissues with a mirVana^®^ miRNA Isolation Kit (Invitrogen, Carlsbad, CA, USA) as recommended by the microarray manufacturer. RNA extraction was performed using the Trizol (Invitrogen, Carlsbad, CA, USA) method for qPCR expression analysis.

### 2.3. Microarray Hybridization

The miRNA profile was analyzed with a GenChip miRNA 3.0 array (Affymetrix, Santa Clara, CA, USA); this chip contained 1733 human mature miRNA probe sets. For the analysis, 500 ng of RNA was tailed with poly(A) and labeled with biotin using the FlashTag™ Biotin HSR RNA Labeling kit (Applied Biosystems™, Vilnius, Lithuania). Hybridization and washing of bio-labeled RNA samples were performed with Affymetrix GeneChip Fluidics Station 450 (Affymetrix, Applied Biosystems™, Santa Clara, CA, USA), following the manufacturer’s instructions. Finally, the microarrays were scanned with the Affymetrix GeneChip Scanner 3000 (Affymetrix, Santa Clara, CA, USA).

### 2.4. Determination of miRNAs with Differential Expression

To assess the differential expression of miRNAs between healthy vs. LACC tissues, datasets were screened with the Limma package (Version 3.57.0) in the R environment (Version 4.3.0) [22]. For validation, we only considered those miRNAs with differential expression higher than 2-fold with adjusted *p* value < 0.01.

### 2.5. MiRNA Targets Prediction and KEGG Enrichment Analysis

The Encyclopedia of RNA Interactomes (ENCORI) database [23] was consulted to identify potential targets for the validated miRNAs. Default parameters were used to identify miRNA-mRNA interactions, with the support of Ago CLIP-seq data experiments and at least one target-predicting program (PITA, RNA22, miRmap, DIANA-microT, miRanda, PicTar, and TargetScan). Then, all target gene names were compiled to perform a pathway enrichment analysis of gene ontology (GO).

For the GO assay, the Kyoto Encyclopedia of Genes and Genomes (KEGG) pathway enrichment was carried out using WEB-based GEne SeT AnaLysis Toolkit (WebGestalt, Version 2019) [24]. The parameters used were as follows: Organism: *Homo sapiens*, method: over-representation analysis (ORA), functional database: pathway, enrichment categories: wikipathway and wikipathway cancer.

### 2.6. Cell Culture and Transfection

Human cervical cancer-derived cell lines, C-33A, CaSki, HeLa, and SiHa, were cultured in Dulbecco’s modified Eagle’s medium F12 MEDIUM (DMEM/F-12) (Gibco, Grand Island, NY, USA) supplemented with 10% fetal bovine serum (Gibco, Grand Island, NY, USA) and maintained at 37 °C with 5% CO_2_. For comparison purposes, HaCat cells (derived from non-tumoral keratinocytes) were maintained in Dulbecco’s modified Eagle’s medium (DMEM) with 10% fetal bovine serum (Gibco, Grand Island, NY, USA) and 1% antibiotics (Cytiva, Logan, UT, USA). All cell lines were obtained from ATCC. The miRNA mimics, inhibitors, and plasmids were transfected or co-transfected using a Lipofectamine 3000 Transfection Kit (Invitrogen, Carlsbad, CA, USA) following the manufacturer’s instructions. RNA expression analysis was performed 24 h post-transfection.

### 2.7. RNA Expression Analysis

The expression of mature miRNAs and genes of interest was performed by real-time PCR. We employed 100 ng of RNA and TaqMan probes (Applied Biosystems, Pleasanton, CA, USA) to synthesize specific cDNA for each miRNA. Following the manufacturer’s recommendations, a TaqMan Micro-RNA Reverse Transcription Kit (Applied Biosystems, Vilnius, Lithuania) was employed for cDNA synthesis and qPCR was performed with 1 μL of cDNA and TaqMan™ Universal Master Mix II, no UNG (Applied Biosystems, Vilnius, Lithuania). For mRNA quantification, cDNA was synthesized using 1000 ng of total RNA using the High-Capacity cDNA Reverse Transcription Kit (Applied Biosystems, Vilnius, Lithuania). Specific primers (Appendix A) and PowerUp™ SYBR™ Green Master Mix (Applied Biosystems, Vilnius, Lithuania) were used for the qPCR test. All qPCRs were performed using the StepOne instrument (Applied Biosystems, Singapore). Relative expression was calculated with the 2^−ΔΔCT^ method using RNU6B and β-actin expression levels as endogenous control.

### 2.8. Luciferase Reporter Assays

The putative miRNA-binding sites of JAK1 and STAT3 (wild-type or mutant) were cloned into pmiR-report plasmid (Ambion, Vilnius, Lithuania). These vectors were co-transfected with mimic (to overexpress) or antimiR (to knockdown) for miR-34c or miR-30a in HeLa cells using Lipofectamine 3000 (Applied Biosystems, Invitrogen, Carlsbad, CA, USA). Then, 24 h after transfection, luciferase activity was quantified using the Dual-Luciferase Reporter Assay System (Promega, Madison, WI, USA) and GloMax 96 Microplate Luminometer (Promega, Sunnyvale, CA, USA). Firefly luciferase activity was normalized to Renilla activity; each experiment was performed in triplicate. Cells transfected with miR-1, scrambled sequences, or non-transfected cells were used as controls.

### 2.9. Cell Viability Assays

Briefly, 2500 cells were seeded in a 96-well plate. After 24 h incubation, they were transfected with miRNA mimic, antimiR, or control. Cell viability at 24, 48, or 72 h was then determined through the MTT assay. Cells were washed with PBS and exposed to MTT (Sigma-Aldrich, St. Louis, MO, USA) for 3 h at 37 °C; then, they were washed and incubated with 100 μL DMSO for 15 min. The optical density (OD) was recorded at 540 nm in an Epoch microplate spectrophotometer (Biotek, Winooski, VT, USA).

Cell viability and proliferation were also measured using Cell Titer-Glo luminescent cell viability assay (Promega) 72 h post-transfection, following the manufacturer’s instructions.

### 2.10. Colony Formation Assay

After 24 h, pre-transfected HeLa cells were cultured and seeded in 6-well plates (500 cells/well). After ten days of culture, cells were fixed with 6% glutaraldehyde for thirty minutes and stained with 1% crystal violet for ten minutes. The colonies were counted using the ImageJ software (Version 1.54i).

### 2.11. Statistical Analysis

Data are expressed as the mean ± the standard deviation of three independent experiments. P values less than 0.05 were considered statistically significant and are represented as *p* < 0.05 *, *p* < 0.01 **, and *p* < 0.001 ***. All statistical analyses were performed in the GraphPad Prism 6 software.

## 3. Results

### 3.1. Clinicopathological Characteristics of Patients with LACC

For this study, a total of 46 patients diagnosed with LACC were enrolled, accepted, and signed to confirm informed consent. From them, 31 samples were assessed for miRNA profile expression using microarray assays, and the other 15 samples were used for validation. The mean age was 40, ranging from 29 to 45 years. According to the FIGO criteria staging classification, most patients were classified as IIB (69%) or IIIB (26%). The histologic subtypes were squamous cell carcinoma (87%) and adenocarcinoma (13%). All patients were tested for HPV infection, and the most prevalent genotypes were HPV-16 (47.8%), HPV-18 (26.1%), and HPV-45 (17.4%). Finally, according to RESIST treatment response guidelines, 47% of patients were categorized as complete response (CR) and 52% as non-response, including patients with partial response (PR), progressive disease (PD), and stable disease (SD). Table 1 summarizes the described clinicopathological characteristics.

### 3.2. A miRNA Expression Profile Differentiated between LACC and Healthy Tissues

We compared miRNA expression profiles between LACC and healthy tissues using the Affymetrix GeneChip™ miRNA 3.0 arrays. The differential expression for each miRNA was determined using the LIMMA package in R and the Euclidean distances method. A total of 615 miRNAs displayed differences in their expression levels (adjust *p*-value < 0.01). This profile is represented in the hierarchical heat map in Figure 1; 271 miRNAs showed high expression levels (plotted in red), whereas 344 miRNAs exhibited low expression levels (plotted in blue) compared with healthy tissue. Color intensity represents the expression levels for each miRNA relative to the mean expression value.

### 3.3. Validation of miRNAs Expression by qPCR

For validation, we selected nine miRNAs with a fold change higher than 2, an adjusted *p*-value < 0.01 obtained from the microarray data, and with previous reports of dysregulation in cervical cancer (Table 2).

The expression of these nine miRNAs was assessed by qPCR in the independent cohort of 15 LACC patients (see above). Only seven analyzed miRNAs maintained differential expression levels with statistical significance (*p* < 0.01) in LACC tissues: miR-34b-5p, miR-30a-5p, miR-10a-5p, miR-34c-5p, miR-375-3p, and miR-486-5p were downregulated, while only miR-196a-5p was upregulated (Figure 2). These results confirmed the data obtained in the microarray assays.

### 3.4. Differentially Expressed Validated miRNAs Regulate JAK/STAT Signaling Pathways Components

The target genes of the previously validated miRNAs were identified using the ENCORI database. This database comprises miRNA-mRNA interactions experimentally validated with the Ago CLIP-seq test [23]. A total of 9918 mRNAs (including repeated targets among miRNAs) could be regulated by these miRNAs. The genes targeted by each miRNA are listed in Appendix A. Later, to predict the impact of this miRNA group on biological pathways, we performed an overrepresentation analysis (ORA) using the WebGestalt database. The Wikipathway functional database analysis yielded leptin, androgen receptor, gastrin, and TGF-beta signaling pathways as the most represented (Figure 3a). To focus the analysis on cancer-related signaling pathways, we used the Wikipathway cancer database to perform a new enrichment analysis. PDGFR-beta, IL-6, IL-1, non-small cell lung cancer, and TGF-beta signaling pathways were the most enriched (Figure 3b).

Our group previously reported that JAK-STAT pathway components were overexpressed [18,38], and high levels of these proteins were also observed in LACC tissues (Figure 4) [39].

In line with this evidence, we decided to explore whether elements of this pathway were regulated by the differentially expressed miRNAs we found using the ENCORI database [23]; we found that miR-30a and miR-34c could potentially regulate JAK1 and STAT3, respectively, with the highest probability among the analyzed miRNAs. Previous data from our group had shown that IL6R, JAK1/JAK2, and STAT3/STAT5b were dysregulated in locally advanced cervical cancer (Appendix A) [18].

### 3.5. The Expression of miR-30a and miR-34c Was Negatively Correlated with the Expression of JAK/STAT Genes in CC-Derived Cell Lines

To demonstrate the regulation of miR-30a and miR-34c on the JAK/STAT genes, we analyzed the basal expression of miRNAs and their targets in C-33A, CaSKi, HeLa, and SiHa CC-derived cell lines. As shown in Figure 5a,b, the expression levels of both analyzed miRNAs were lower by 0.5-fold in C33A and CaSKi cells and at least 0.8-fold in HeLa and SiHa cell lines compared with their expression in non-transformed HaCat cells. In contrast, the levels of JAK/STAT genes JAK1 and STAT3 were significantly overexpressed in all analyzed CC cell lines relative to HaCat cells (Figure 5c,d). These results correlated with our findings in LACC tissues.

### 3.6. JAK1 and STAT3 Are Directly Regulated by miR-30a and miR-34c Respectively and Its Transcriptional Targets

To predict the interaction of miR-30a/JAK1 and miR-34c/STAT3, we used TargetScan to identify the miRNA-mRNA binding sites. Figure 5e,f show the interaction sequences between the specific 3′UTR of the mRNA and the miRNA. Wild-type (WT) or mutated (MUT) 3′UTR interaction sites from the JAK1 or STAT3 mRNAs were cloned in the pmiR-Report reporter system to validate their specific interaction with the corresponding miRNAs (Figure 5e,f). As shown in Figure 5g,h, luciferase activity was lower when JAK1-WT and STAT3-WT were co-transfected with miR-30a and miR-34c mimics, respectively. Conversely, when the miRNA levels were knocked down with the corresponding antimiRs, the luciferase activity was higher only for miR-34c/STAT3 interaction. No reporter activity changes were observed when the MUT sequence of each mRNA was co-transfected with mimics or antimiRs for each miRNA.

Finally, to demonstrate the impact of miR-30a and miR-34c on the transcriptional activity of the JAK-STAT signaling pathway, we measured the expression of two transcriptional targets of the pathway, CXCL10 and CCND1. As shown in Figure 5i, when the miR-34c mimic was transfected, the relative expression of CXCL10 and CCND1 was significantly decreased. Moreover, miR-30a mimic was able to downregulate only CCND1. Altogether, these data demonstrated the direct regulation exerted by miR-30a and miR-34c on the JAK1 and STAT3 mRNAs, respectively.

### 3.7. miR-30a and miR-34c Overexpression Decreased Viability and Proliferation in HeLa Cells

To explore the biological function of miR-30a and miR-34c expression, cell viability was tested with MTT assays in the HeLa cell line (Figure 6a). The overexpression or knockdown of the miRNA was induced by transfection of mimic or antimiR, respectively. The results showed that 48 h post-transfection, the cells expressing high levels of miR-30a and miR-34c decreased their viability by 30% compared with controls or those with the knocked-down miRNA. This effect was maintained for up to 72 h. To confirm this finding, we assessed the cell proliferation with the CellTiter-Glo assay in the same cell line (Figure 6b). We observed a similar effect 72 h post-transfection; when overexpressing miR-30a and miR-34c, the relative luminescence decreased, suggesting lower cell viability and proliferation than control conditions.

Next, we performed a colony-formation assay to confirm the biological effect of miR-30a and miR-34c in cervical cancer. Figure 6c shows that overexpression of miR-30a and miR-34c in the HeLa cell line significantly reduced the number of colonies compared with controls. This evidence supports the negative effect of miR-30a and miR-34c on cell viability and proliferation.

## 4. Discussion

In this study, we established a profile of 615 differentially expressed miRNAs in young (<45 y) women with locally advanced cervical cancer (LACC). We validated a group of seven miRNAs in an independent cohort: one of them—miR-196a-5p—was upregulated, and six—miR-34b-5p, miR-30a-5p, miR-10a-5p, miR-34c-5p, miR-375-3p, and miR-486-5p— downregulated. On performing pathway enrichment analysis with the target genes for these seven miRNAs, we found overrepresented genes of the JAK-STAT pathway, among others. Two of the miRNAs we found significantly downregulated in LACC—miR-30a and miR-34c—are direct regulators of JAK1 and STAT3. Both microRNAS significantly reduced expression levels of signaling pathway target genes and decreased HeLa cell line proliferation when their expression was restored.

All the seven miRNAs we validated have been found by other groups to vary their expression in cervical cancer. Our findings regarding five of them correspond to these previous reports; miR-196a-5p was consistently upregulated [26], while miR-34b-5p [30] and miR-34c-5p [33], miR-30a-5p [31], and miR-375-3p [25] were consistently downregulated. We found miR-10a-5p and miR-486-5p to be downregulated, in contrast to reports by Pereira [25] and Li [37], respectively. Other groups have reported miR-10a-5p downregulation in CC-derived cell lines [32], consistent with our results. Still, its expression in tumors might be as variable as in colorectal cancer, where its expression distinguished groups’ responses to therapy [40]. This discrepancy in miR-10a-5p expression suggests that other regulation factors not considered in this study are at play; further research on miR-10a-5p will determine its role in cervical cancer. Regarding miR-486-5p, we found only one report of its overexpression in cervical cancer [37]; it is regarded as a PTEN-negative regulator in different cancer types, such as lung cancer and myeloid leukemia [41]. So, the role of miR-486-5p in cervical cancer warrants further investigation.

The seven under-expressed miRNAs that we found regulate the JAK-STAT pathway through several of its elements. This supports previous findings on overactivation of the JAK-STAT pathway in LACC [18,42,43,44,45]. In addition, indirect evidence of JAK-STAT overactivation is available through studies that analyze the inflammatory effectors it regulates. For instance, IL-1β is a marker of intraepithelial lesions in young women [46], and CXCL10 expression is a predictor of squamous cell carcinoma across all age groups [47].

Other studies have shown overexpression of genes directly or indirectly related to inflammation, such as the transcriptomic analysis of peripheral blood mononuclear cells (PBMCs) from patients with cervical cancer (CC) versus cervical intraepithelial neoplasia (CIN) and healthy tissues by Ndiaye et al. [48], where IL1R2 and IL18R1 were significantly overexpressed. High levels of inflammatory cytokines have also been associated with persistent HPV infectionl [15,49], which is consistent with increased JAK-STAT overactivation. STAT3 was implicated in the progression from HSIL to SCC in a report that also found that the upregulation of several genes involved in immune response and inflammation was crucial in the transformation from normal cervical cells to low-grade squamous intraepithelial lesions (LSIL) [16]. Likewise, Domeneci et al. reported better prognosis in CC patients with lower IL-6 expression in a population with an average age of 45 [50]. Moreover, it has been demonstrated that overactivation of JAK-STAT signaling favors immune escape through signaling targets such as PDL1, a negative regulator of immune response [51]. Together with independent observations of an association between inflammation and a worse prognosis for cervical cancer patients [52], these reports highlight the involvement of pro-inflammatory signaling in the early-stage development of cervical cancer.

Consequently, the downregulation of miRNAs that target elements of the JAK-STAT pathway might provide early insight into the inflammatory status of cervical tumors and its impact on the development of cervical cancer in young women. This hypothesis is supported by works that show a direct effect of HPV oncoproteins on the expression of several miRNAs [53,54]; in particular, Chiantore and collaborators [53] found that HPV E6 decreased miR-34a expression, most likely through p53 regulation shared with the miR-34 family that includes miR-34b-5p and miR-34c-5p, reported in the current work.

We focused only on miRNAs that target the JAK/STAT pathway, due to our interest in cancer-related inflammation, but the participation in CC development of remaining miRNAs with validated differential expression remains to be determined. Interestingly, other possible targets of the seven validated miRNAs are related to the cancer hallmarks of apoptosis evasion, genomic instability, sustained angiogenesis, and invasion and metastasis (see Appendix A), which are fundamental to cervical cancer progression. Previous reviews have explored the important relationship between deregulated miRNAs and CC and their involvement in tumor progression [55,56].

Most cervical cancer genomic studies have sourced data from patients encompassing the whole age range. For instance, Castanon et al. found significantly worse survival in younger women diagnosed with early-stage cervical cancer and no difference among age groups in women diagnosed with locally advanced cervical cancer [57]; on the other hand, Isla-Ortiz and colleagues concluded that age at diagnosis made no difference in cervical cancer prognosis [6]. However, there is evidence that contradicts these reports showing differences between age groups: Xie et al. found worse survival in older (65+) patients, both globally and stratified [58], while Lau et al. [7] found a higher metastasis rate in patients aged 30 or younger. Moreover, at the time of writing, studies that purposefully compare miRNA or even ncRNA expression between different age groups are still lacking and, therefore, urgently needed. It is important to continue searching for specific age-related markers, especially those associated with early CC development in younger patients, who are often diagnosed in advanced stages [2,5].

Our study showed that miR-30a and miR-34c are such candidates since they were under-expressed in CC, releasing JAK1 and STAT3 from their regulation and thus increasing proliferation. miR-30a and miR-34c join miR-375 [36,55] as under-expressed ncRNA tumoral biomarkers. Previous research has associated both miRNAs with regulation of proliferation in CC [33,59,60]; however, this is the first report to delve into the role of miR-34c and miR-30a as regulators of inflammation through the JAK-STAT pathway. Interestingly, JAK1 and STAT3, targeted by miR-34c and miR-30a, are currently subject to attention from several research groups; for instance, they have been proposed as therapeutic targets [61]. Further studies are needed to establish miR-30a and miR-34c as bona fide markers, including searching for an association between miR-30a and miR-34c and overall survival in a larger number of patients, overcoming the current study’s limitations, and functional demonstrations of their function using mimics or antimiRs. Studies that provide deeper detail on the pro-inflammatory function of miR-30a and miR-34c will complete the picture. Given that CC still affects women of all ages and that the median age for diagnosis is 50 years [4], efforts towards earlier diagnosis are still necessary. 

## Figures and Tables

**Figure 1 cells-13-00896-f001:**
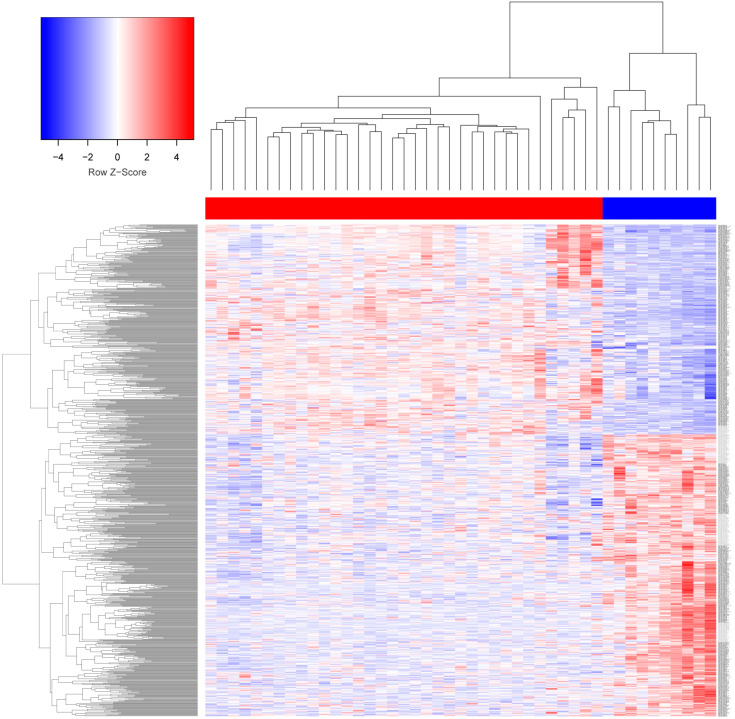
Hierarchical cluster from 31 LACC (red bar) and 10 healthy tissue (blue bar) samples. The heatmap shows 615 miRNAs differentially expressed (271 overexpressed miRNAs and 344 under-expressed miRNAs). The relative abundance of each miRNA is represented in z-score values, and the intensity of the color is correlated to the level of expression (red, over-expressed; blue, downregulated; white, no change).

**Figure 2 cells-13-00896-f002:**
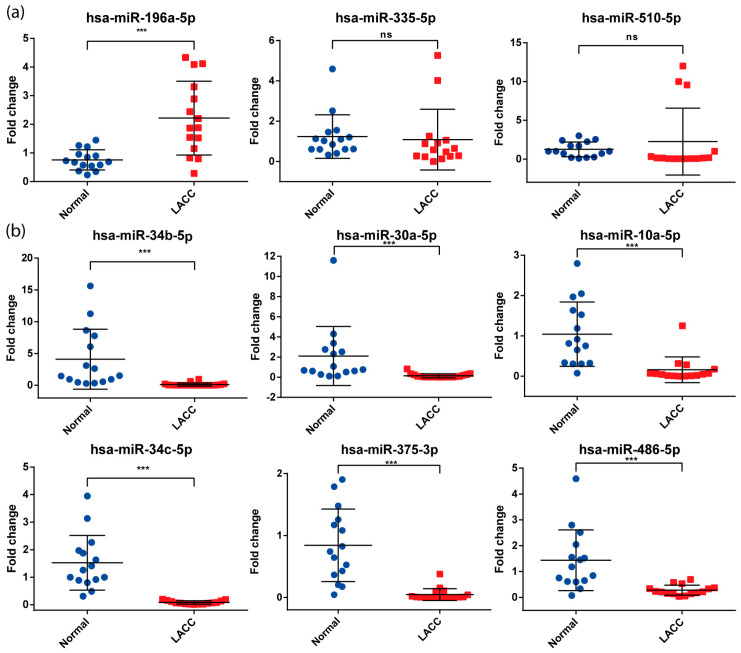
miRNAs validated using qPCR in an independent cohort of 15 patients with LACC versus normal tissue. (**a**) miRNAs overexpressed in LACC samples and (**b**) miRNAs underexpressed in LACC samples. *** *p* < 0.001; ns = no significance.

**Figure 3 cells-13-00896-f003:**
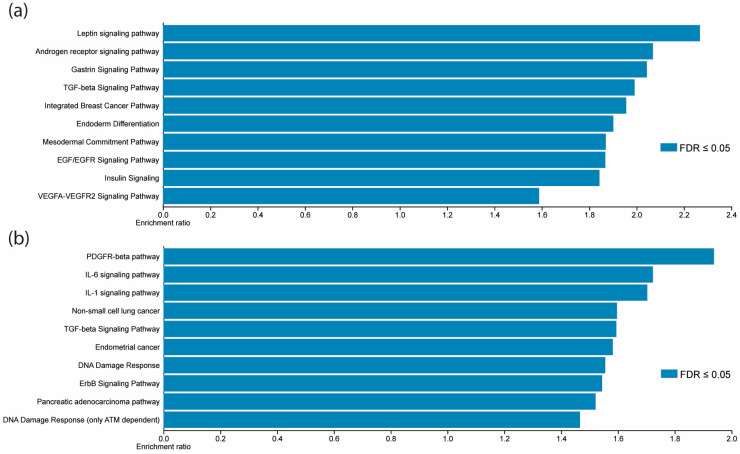
The seven different expressed miRNAs validated in LACC are associated with inflammation-related pathways. (**a**) Leptin, androgen receptor, gastrin, and TGF-beta signaling pathways were the most enriched according to the Wikipathway database. (**b**) PDGFR-beta, IL-6, and IL-1 were the most enriched according to the Wikipathway cancer database. The *x*-axis represents the enrichment ratio.

**Figure 4 cells-13-00896-f004:**
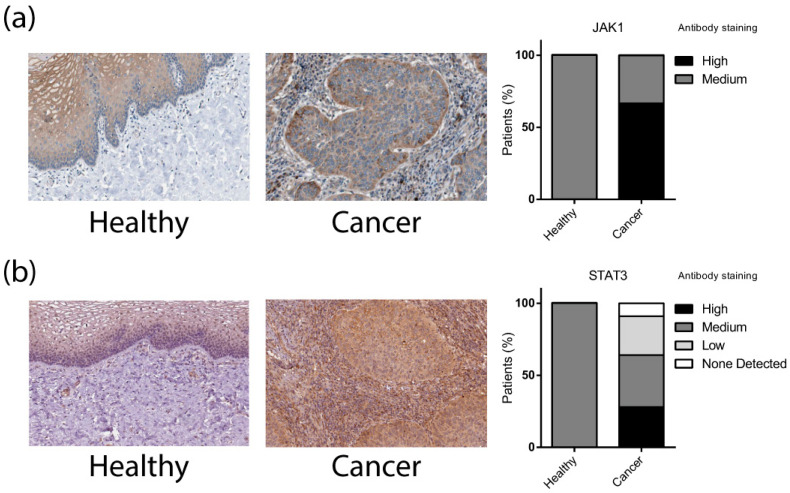
Immunohistochemical analysis of JAK1 and STAT3 proteins overexpressed in CC according to The Human Protein Atlas. (**a**) JAK1 immunohistochemistry in healthy and cervical cancer tissue; (**b**) STAT3 immunohistochemistry in healthy and cervical cancer tissue. The bar graph represents the percentage of patients based on the intensity of the antibody staining. All data were obtained from The Human Protein Atlas version 23.0 https://www.proteinatlas.org/ (Accessed on 9 December 2023).

**Figure 5 cells-13-00896-f005:**
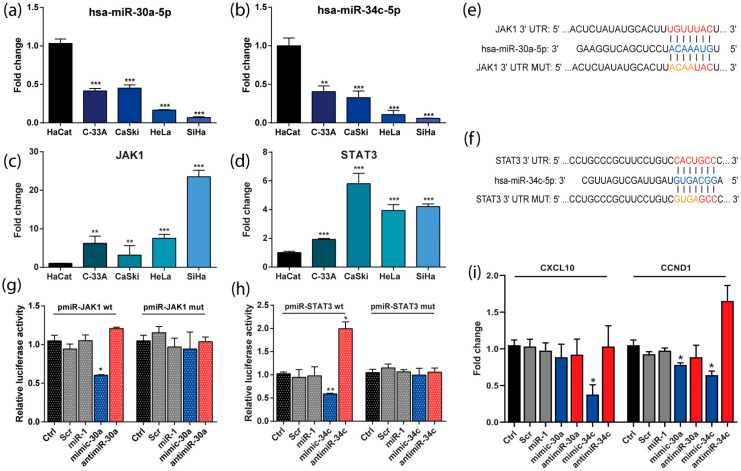
miR-30a and miR-34c regulated the JAK-STAT signaling pathway and its transcriptional targets. (**a**,**b**) miR-30a and miR-34c relative expression in CC-derived cell lines; (**c**,**d**) JAK1 and STAT3 relative expression in CC-derived cell lines; (**e**) binding site of miR-30a on 3′UTR JAK1 and (**f**) miR-34c on 3′UTR STAT3 predicted through bioinformatics (The red letters represent the 3’UTR wild type region of the mRNA, the orange letters the mutated region and the blue letters the seed region of the miRNA); (**g**) luciferase activity of miR-30a/JAK1/JAK1mut interaction and (**h**) miR-34c/STAT3/STAT3mut in HeLa cells; (**i**) effects of miR-30a and miR-34c on Jak/STAT expression targets CXCL10 and CCND1. * *p* < 0.05, ** *p* < 0.01, *** *p* < 0.001.

**Figure 6 cells-13-00896-f006:**
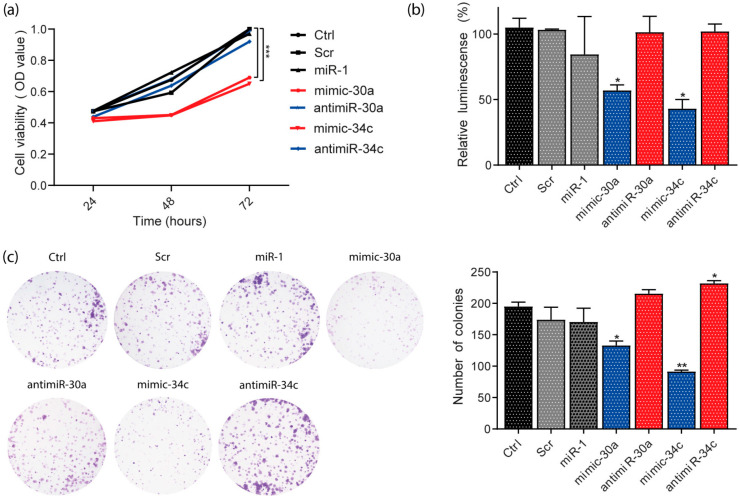
miR-30a and miR-34c regulate viability and proliferation in the HeLa cell line. (**a**) Viability of HeLa cells 24, 48, and 72 h post-transfection of miR-30a and miR-34c, evaluated using MTT; (**b**) proliferation of HeLa 72 h post-transfection of miR-30a and miR-34c measured with CellTiterGlo; (**c**) cell survival determined via colony formation assay. * *p* < 0.05, ** *p* < 0.01, *** *p* < 0.001.

**Table 1 cells-13-00896-t001:** Patients’ clinical and histological characteristics.

Characteristics	*N*	%
Age (mean, standard deviation)	40 (±5)	
Clinical stage		
IB2	2	4.34%
IIB	32	69.56%
IIIB	12	26.08%
Histological type		
Adenocarcinoma	6	13.04%
Squamous cell carcinoma	40	86.95%
Tumor size		
<4 cm	6	13.04%
>4 cm	40	86.95%
HPV Genotyping		
Type 16	22	47.8%
Type 18	12	26.1%
Type 45	8	17.4%
Other	4	8.7%
Treatment outcome		
Complete (RC)	22	47.82%
No response (PR, PD, SD)	24	52.17%

**Table 2 cells-13-00896-t002:** miRNAs selected for their fold change, adjusted *p*-value, and with previous reports, for validation in the independent cohort of LACC patients.

miRNA	Fold Change	Adj *p*-Value	Previous Reports
hsa-miR-196a-5p	3.07	2.84 × 10^−5^	Pereira, et al., 2010 [25]; Villegas-Ruiz, et al., 2014 [26]
hsa-miR-335-5p	2.63	7.17 × 10^−7^	Wang and Jiang, 2015 [27]
hsa-miR-510-5p	2.12	3.15 × 10^−8^	Yoshida, et al., 2021 [28]
hsa-miR-34b-5p	−2.39	1.01 × 10^−5^	Jiménez-Wences, et al., 2016 [29]; Cao, et al., 2019 [30]
hsa-miR-30a-5p	−2.75	4.52 × 10^−5^	Bayramoglu et al., 2021 [31]
hsa-miR-10a-5p	−3.00	6.97 × 10^−6^	Pereira, et al., 2010 [25]; Zhai et al., 2017 [32]
hsa-miR-34c-5p	−3.52	6.90 × 10^−6^	Jiménez-Wences, et al., 2016 [29]; Wei, et al., 2021 [33]
hsa-miR-375-3p	−5.12	1.13 × 10^−7^	Li, et al., 2011 [34]; Wilting, et al., 2013 [35]; Tiang, et al., 2014 [36]
hsa-miR-486-5p	−5.69	4.12 × 10^−12^	Li et al., 2018 [37]

## Data Availability

All data are available on request to Oliver Millan-Catalan (oliver.millan.sg@gmail.com) and Carlos Pérez-Plasencia (carlos.pplas@gmail.com).

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
