# Peer review of "A microRNA Profile Regulates Inflammation-Related Signaling Pathways in Young Women with Locally Advanced Cervical Cancer"

_cells, 2024, doi:10.3390/cells13110896_

Round 1

Reviewer 1 Report

Comments and Suggestions for Authors

The authors present a systematic study that validates the involvement of inflammatory pathways, specifically the JAK/STAT pathways in HPV mediated cervical cancer and offers diagnostic targets for early detection in younger patients. As such this is a fairly defined scope of interest but I appreciate the various modes of validation employed by the authors in the study.

Below are my specific comments;

1. Given the smaller number of test and control patients, can you add some commentary on the power of your study and any impact of the stratification shown in Table 1?

2. The rationale for selection of the 9 miRNAs from the 615 differentially expressed ones seems a little weak based on the criteria described. Were these the only ones that had a 2 or greater than 2 fold change? The chart seems to show otherwise. Also, what reports are you referring to in the selection? I would expect some references here. Please describe this section a bit better.

3. In a detection scheme, presence or increase of a target is always a better evaluation than looking for something that is repressed. How would you propose these miRNAs be used as diagnostic markers is they are under-expressed relative to normal tissue?

4. While a majority of the miRNA targets were mentioned as being involved in the JAK/STAT pathway, can the authors comment on what other relevant targets of interest could help explain or advance the mechanism of progression of cervical cancer in younger patients compared to older ones? It seems a shame to not address the 606 other miRNAs that were seen to be differentially expressed and perhaps break them into tiers of fold change and report.

Author Response

The authors present a systematic study that validates the involvement of inflammatory pathways, specifically the JAK/STAT pathways in HPV mediated cervical cancer and offers diagnostic targets for early detection in younger patients. As such this is a fairly defined scope of interest but I appreciate the various modes of validation employed by the authors in the study.

Below are my specific comments;

  1. Given the smaller number of test and control patients, can you add some commentary on the power of your study and any impact of the stratification shown in Table 1?

Dear Reviewer, we appreciate your positive thoughts about our work. Statistical power analysis focuses more on the probability that a hypothesis is true. In our study, we do not assume a comparative hypothesis between younger cancer patients and older patients. It was simply a descriptive study of 46 patients whose average age was 40 years. Although the question is quite important, when comparing two populations or when there is a concrete hypothesis in the study, for descriptive studies it is not possible to determine the power of the analysis.

Regarding Table 1, the inclusion criteria were patients with locally advanced cervical cancer; most of the patients recruited were stratified in Stage IIB. There was no preselection for that stage. In addition, the majority was Squamous Cell Carcinoma as reported in previous studies.

  1. The rationale for selection of the 9 miRNAs from the 615 differentially expressed ones seems a little weak based on the criteria described. Were these the only ones that had a 2 or greater than 2 fold change? The chart seems to show otherwise. Also, what reports are you referring to in the selection? I would expect some references here. Please describe this section a bit better.

Dear reviewer we apologize for not being adequately clear. The first selection criteria from the list of 615 miRNAs whose p-value was <0.01; was the fold-change( > 2, <-2). As the values are logarithmic in base 2; this filter allows us to select those microRNAs with more than 4-fold over- or under- expressed with respect to normal tissues. The next criterion was that the selected molecules had been reported in studies involving patients with cervical cancer. To focus on the analysis of a small number of molecules. These works are now listed in Table 2.

  1. In a detection scheme, presence or increase of a target is always a better evaluation than looking for something that is repressed. How would you propose these miRNAs be used as diagnostic markers is they are under-expressed relative to normal tissue?

Since biomarkers are measurable characteristics and not molecules themselves, the absence of a given molecule is as valid as its presence. One of the most conspicuous examples is the triple-negative breast cancer, which lacks ER, PR, and HER2; additionally, several recent works report such cases: Won et al, reported an association between a loss of INPP4B and basal-like breast cancer (DOI: 10.1038/modpathol.2013.97); Ashizawa et al, proposed negative expression of ARID1A as an early-stage prognostic biomarker for undifferentiated gastric cancer (DOI: 10.1038/s41598-019-43293-5); and Roa-Peña et al, identified K17 as a novel negative prognostic biomarker for pancreatic cancer (DOI:10.1038/s41598-019-47519-4). Furthermore, the down-regulation of ncRNAs has also been reported as biomarkers, e.g., MT1JP, as reported by Yang et al. (DOI 10.1089/cbr.2019.3328). And specifically in CC, under-expressed of miR-375 as reported by tiang et al (doi: 10.1093/jnci/dju241). The text has been modified (Line 416-419, Discussion section).

  1. While a majority of the miRNA targets were mentioned as being involved in the JAK/STAT pathway, can the authors comment on what other relevant targets of interest could help explain or advance the mechanism of progression of cervical cancer in younger patients compared to older ones? It seems a shame to not address the 606 other miRNAs that were seen to be differentially expressed and perhaps break them into tiers of fold change and report.

Dear reviewer, we understand your concern in suggesting that it is necessary to understand the role of all deregulated microRNAs. However, a single microRNA can regulate thousands of canonical genes (doi:10.1182/blood-2006-01-030015). So it could be extremely complex to describe these regulatory networks. For this purpose, we made a figure (Supplementary Figure 1) in which we analyzed the impact of these seven characterized microRNAs on cancer-related pathways. In the Discussion section we added a paragraph (lines 395-402) with the aim of deepening the knowledge of these seven microRNAs described in the present study.

Reviewer 2 Report

Comments and Suggestions for Authors

The manuscript by Oliver M C et al addresses the investigation of locally advanced cervical cancer (LACC) in women under 45 years. This study is designed based on tissue samples according to clinicopathological characteristics, in vitro and in silico analysis. Here, the authors investigate the differential miRNA profiling between healthy and LACC subjects. In addition, the regulation by the miRNA target genes related to signaling pathways linked to cancer is also studied.

Major concerns

The authors performed a study design shown in the manuscript titled “A microRNA profile regulates inflammation-related signaling pathways in young women with locally advanced cervical cancer”. However, there are some concerns that the authors should consider.

In abstract section, the authors describe that JAK/STAT signaling as a possible diagnostic marker for CC-patients younger than 45 years, but this manuscript does not perform a comparative study between patients >45 years vs <45 years.

In lines 260 - 264, the bibliographic supports (Refer. 18 and 23) do not indicate the information described by the authors. In reference 18, IL6, JAK1/JAK2, and STAT3/STAT5b are not indicated as potential targets of the miRNAs proposed in the manuscript. In reference 23, higher probability of regulation of miR-30c and miR-34c is not documented. Therefore, the selection of further study including miR-30c and miR-34c and excluding miR-34b-5p, miR-10a-5p, miR-375-3p and miR-486-5p is missing and should be detailed.

The authors refer to study the signaling pathways related to inflammation such as JAK/STAT. However, this pathway is already known in relationship with cancer. The authors should demonstrate a molecular marker of this signaling or a marker of cervical cancer when the regulation by mimic or antimiR is performed. Signaling pathway of JAK/STAT is related to broad biological processes.

Minor concerns

Why did authors perform two methods of RNA purification? This methodology leads to possible bias caused by differential performance in isolating RNA from tissue samples.

Author Response

Reviewer 2

Major concerns

The authors performed a study design shown in the manuscript titled “A microRNA profile regulates inflammation-related signaling pathways in young women with locally advanced cervical cancer”. However, there are some concerns that the authors should consider.

In abstract section, the authors describe that JAK/STAT signaling as a possible diagnostic marker for CC-patients younger than 45 years, but this manuscript does not perform a comparative study between patients >45 years vs <45 years.

Dear reviewer, we appreciate your kind comments. As you pointed out, this work is not a comparative study but rather a descriptive approach. We consider that reports concerning miRNA expression in younger (<45 y) patients are scarce in the literature. The text has been modified to clarify our approach and to point out the need for further comparative studies (Lines 40-41 Abstract Section and 411-415 Discussion Section).

In lines 260-264, the bibliographic supports (Refer. 18 and 23) do not indicate the information described by the authors. In reference 18, IL6, JAK1/JAK2, and STAT3/STAT5b are not indicated as potential targets of the miRNAs proposed in the manuscript. In reference 23, higher probability of regulation of miR-30c and miR-34c is not documented.  

Reference 18 (Campos-Parra et al., 2016) shows overexpression of Jak2 and other members of the JAK/STAT pathway, not their relationship with miRNAs; our rationale was to study miRNAs targeting these genes. There was an honest mistake in Reference 23, it was meant to cite the ENCORI database paper (Li et al., 2013); we used ENCORI’s web portal to analyze the probability of miR-30a and miR-34c binding to its targets. We apologize for the confusion and have modified the text accordingly (Line 268-272). 

As for the selection of miR-30c and miR-34c, it was based on their targeting elements of the JAK/STAT pathway, evident in the raw data of the microarrays on which Reference 18 (Campos et al., 2016) is based. The text has been modified to reflect this, mentioning previous, non-shown data from our group; the data is available should the reviewers or editors deem it necessary (lines 272-274, Results Section).

The authors refer to study the signaling pathways related to inflammation such as JAK/STAT. However, this pathway is already known in relationship with cancer. The authors should demonstrate a molecular marker of this signaling or a marker of cervical cancer when the regulation by mimic or antimiR is performed. Signaling pathway of JAK/STAT is related to broad biological processes.

Thank you for this insightful comment. As discussed in point one, this is a descriptive report. We consider it relevant given the lack of reports concerning miRNA expression in younger (<45 y) patients. However, functional assays, such as that suggested by the reviewer, are necessary to fully establish the low expression of miR-30c and miR-34c as molecular markers in young cervical cancer patients; the text has been modified to reflect this more clearly (lines 424-430, Discussion Section). So, for the demonstration of specific markers of Jak/STAT pathway, we performed a qPCR to evaluate the expression levels of two target genes of the pathway. As expected, when miRNA expression is restored, the levels of two target genes are significantly decreased (Fig 5i, lines 307-313).

Minor concerns

Why did authors perform two methods of RNA purification? This methodology leads to possible bias caused by differential performance in isolating RNA from tissue samples.

While undoubtedly possible, this is not the case. The results presented in this and previously published papers from our group (DOI: 10.1016/j.ygyno.2016.07.093, DOI: 10.3390/curroncol29010023) show correspondence between microarray and qPCR data.

Reviewer 3 Report

Comments and Suggestions for Authors

This is an interesting and well-organized article. The authors have to modify their manuscript.

1)      I strongly recommend the authors provide a graphical abstract at the end of the introduction.

2)      It should be noted whether these 45 women with advanced cervical cancer (line 94) started their treatment or not. Because chemotherapy (and any kind of anti-cancer therapy) can change the miRNA profile, which can be different from the untreated person. On the other hand, if the authors profile the miRNA in patient #1 before treatment and patient #2 after two doses of treatment, patient #3 is in the middle of the treatment. Their results cannot be compared.

3)      It would be good if the authors provided the miRNA concentration (table 2) before (healthy) and after cancer (LACC patients). Then “fold change” becomes meaningful.

4)      Did the authors draw Figure 2 based on Table 2?

5)      Is the fold change of miRNAs in Table 2 from the average of 31 CACC and 10 healthy persons?

6)      The authors used Genchip miRNA, which contains 1733 human mature miRNA probe sits. How do they measure the changing of those miRNAs listed in Table 2? On the other hand, did they measure the upregulation and downregulation of 1733 miRNA and find the fluctuations of 9 miRNAs listed in Table 2? Or do they only profile the alterations of the 9 miRNA based on previous reports?

7)      Are the 9 miRNA listed in Table 2 specific for CC disease?

8)      In Figure 6, the description for the bottom-right is missing.

9)      It is very nice if the authors pay attention to the discussion of how changes in miRNAs listed in Table 2 cause invasion in LACC. 

Author Response

Reviewer 3

This is an interesting and well-organized article. The authors have to modify their manuscript.

1)      I strongly recommend the authors provide a graphical abstract at the end of the introduction. EDUARDO

2)      It should be noted whether these 45 women with advanced cervical cancer (line 94) started their treatment or not. Because chemotherapy (and any kind of anti-cancer therapy) can change the miRNA profile, which can be different from the untreated person. On the other hand, if the authors profile the miRNA in patient #1 before treatment and patient #2 after two doses of treatment, patient #3 is in the middle of the treatment. Their results cannot be compared.

Dear Reviewer, thank you very much for your positive feedback.  We totally agree with you on this point. None of the patients included in this study had received previous oncological treatment at the time of biopsy, as mentioned in the Inclusion criteria for patients, point 6 (Line 102-109).

3)      It would be good if the authors provided the miRNA concentration (table 2) before (healthy) and after cancer (LACC patients). Then “fold change” becomes meaningful.

4)      Did the authors draw Figure 2 based on Table 2?

5)      Is the fold change of miRNAs in Table 2 from the average of 31 CACC and 10 healthy persons?

6)      The authors used Genchip miRNA, which contains 1733 human mature miRNA probe sits. How do they measure the changing of those miRNAs listed in Table 2? On the other hand, did they measure the upregulation and downregulation of 1733 miRNA and find the fluctuations of 9 miRNAs listed in Table 2? Or do they only profile the alterations of the 9 miRNA based on previous reports?

Dear Reviewer, we clarify points 3 to 6 in the following text.

 The data in Table 2 is sourced from the microarray data. The fold change was calculated for the 1733 miRNAs using the median expression of 31 CC samples and comparing it to that of the 10 healthy samples using the Limma software (Line 124-127); therefore, no specific concentration data is available.

Then, we selected 9 miRNAs with a fold change higher than 2 and previous reports. Only these 9 miRNAs were  analyzed through qPCR in an independent patient cohort, thus validating the microarray data. The results from this analysis are shown in Figure 2.

The text and table 2 have been modified for clarity.

7)      Are the 9 miRNA listed in Table 2 specific for CC disease?

At the moment, we cannot conclude that these 9 miRNAs are specific. Our data showed they were differentially expressed between tumor and healthy samples and that they target genes belonging to the JAK/STAT pathway.

8)      In Figure 6, the description for the bottom-right is missing.

 Thank you for pointing this out. The figure and its legend have been updated.

9)      It is very nice if the authors pay attention to the discussion of how changes in miRNAs listed in Table 2 cause invasion in LACC. 

The seven microRNAs that were selected from the microarray data regulate different molecular pathways that are hallmarks of cancer. In addition to those related to inflammation, apoptosis evasion, invasion and metastasis and sustained angiogenesis. We made a supplementary figure showing specific genes and hallmarks of cancer that are regulated by these microRNAs (Supplementary figure 1). Finally, we modify the text to enrich the discussion (Line 395-402).

Round 2

Reviewer 2 Report

Comments and Suggestions for Authors

The authors have responded the concerns raised by the reviewer.

Reviewer 3 Report

Comments and Suggestions for Authors

The Authors did not provide a graphical abstract. The rest is OK. MS has been improved.